# Prevalence, Evolution and Prognostic Factors of PASC in a Cohort of Patients Discharged from a COVID Unit

**DOI:** 10.3390/biomedicines13061414

**Published:** 2025-06-09

**Authors:** Mariantonietta Pisaturo, Antonio Russo, Pierantonio Grimaldi, Caterina Monari, Simona Imbriani, Klodian Gjeloshi, Carmen Ricozzi, Roberta Astorri, Caterina Curatolo, Roberta Palladino, Francesco Caruso, Francesca Ambrisi, Lorenzo Onorato, Nicola Coppola

**Affiliations:** 1Department of Mental Health and Public Medicine, University of Campania “Luigi Vanvitelli”, 80138 Naples, Italy; mariantonietta.pisaturo@unicampania.it (M.P.); antonio.russo2@unicampania.it (A.R.); peogrimaldi@icloud.com (P.G.); caterina.curatolo1@gmail.com (C.C.); roberta.palladino@studenti.unicampania.it (R.P.); francescocaruso9190@gmail.com (F.C.); francescambrisi@gmail.com (F.A.); lorenzo.onorato@unicampania.it (L.O.); 2Infectious Diseases Unit, AOU Luigi Vanvitelli, 80131 Naples, Italy; caterina.monari@gmail.com (C.M.); simo.imbriani@gmail.com (S.I.); klodian.gjeloshi@unicampania.it (K.G.); carmenricozzi28@gmail.com (C.R.); 3Department of Medicine and Surgery, LUM University “Giuseppe Degennaro”, 70010 Casamassima, Italy; roberta.astorri@hotmail.com

**Keywords:** COVID-19, Long COVID, prognostic factors, PASC

## Abstract

**Background and Aim**: PASC is a potentially debilitating clinical condition consisting of different general symptoms experienced by about 10% of patients with previous SARS-CoV-2 infection. Our study analyses a cohort of patients with a history of hospitalization for COVID-19 and aims to evaluate prognostic factors for experiencing PASC and to investigate the characteristics of patients experiencing PASC symptoms. **Methods**: This is an observational, monocentric retrospective study including all adult patients admitted to our COVID unit from 28 February 2020 to 30 April 2022, discharged alive, and having performed at least one follow-up visit at our post-COVID outpatient clinic after a minimum of three months from discharge. Patients who experienced persistent clinical manifestations or the development of new symptoms three months after the initial SARS-CoV-2 infection, with these symptoms lasting for at least two months with no other explanation, were defined as having PASC. **Results**: A total of 429 patients were discharged alive from our COVID Unit and 244 patients performed at least one follow-up visit in our outpatient clinic. Of these, 134 patients did not experience PASC, while 110 patients experienced PASC. Long-COVID patients were more frequently female (43.6% vs. 31.3%, *p* = 0.048), more frequently presented throat pain and headache at hospital admission (respectively 8.9% vs. 2.5%, *p* = 0.041 and 15.8% vs. 5%, *p* = 0.007), and were more likely to have a history of type 2 diabetes mellitus (25.5% vs. 13%, *p* =0.013). At the multivariable analysis, female gender, type 2 diabetes, and headache at admission were factors associated with PASC. All 46 patients who performed at least two different admissions in our outpatient clinic were divided in two groups: the first including the 16 patients who experienced a reduction or a resolution of symptoms related to COVID-19, the second comprising the 30 patients who experienced clinical worsening or persisting symptoms. Smoking habit was more represented among patients with stable or worsening symptoms (42.3% vs. 7.7%, *p* = 0.042); myalgias at admission were more frequent in the clinical worsening group (27.6% vs. 0%, *p*= 0.039); and a larger amount of patients who reported neuropsychiatric symptoms and respiratory symptoms were in the stable or worsening PASC symptoms group. **Discussion**: In conclusion, this study underscores the complexity of PASC, identifying female sex, Type 2 diabetes, and certain acute COVID-19 symptoms as potential predisposing factors for its development. PASC still represents a substantial public health challenge, and ongoing efforts are essential to better understand its underlying mechanisms and improve patient outcomes.

## 1. Introduction

Beyond acute infection, well known as COronaVIrus Disease 19 (COVID-19), severe acute respiratory syndrome coronavirus 2 (SARS-CoV-2) is now recognized to be also responsible for a potentially debilitating post-infection multisystem condition defined as long-COVID syndrome, also referred to as post-acute sequelae of SARS-CoV-2 infection (PASC) [1,2,3,4,5]. The World Health Organization describes PASC as occurring in individuals with a history of probable or confirmed SARS-CoV-2 infection, usually 3 months from the onset of COVID-19, with symptoms that last at least 2 months and cannot be explained by an alternative diagnosis [6]

According to the literature, 10% to 50% of convalescent patients are expected to experience one or more symptoms of long-COVID [7,8], with the highest prevalence observed in patients with severe COVID-19 [9] and a lower prevalence with certain SARS-CoV-2 variants, such as Omicron [10,11]. As for the COVID-19 setting, where the presence of certain demographic parameters, conditions, or comorbidities likely increases the risk of evolution in severe disease [12,13,14,15], female sex has been identified as a consistent risk factor for developing PASC, and associations have also been found with smoking habit and a wide range of comorbidities at baseline [1,16,17,18].

However, solid diagnostic criteria for PASC are lacking in the literature and there is no consensus regarding an algorithm of investigation. PASC represents a complex clinical problem since it is highly heterogenous and may be associated with multisystem tissue damage/dysfunction including acute encephalitis, cardiopulmonary syndromes, hepatobiliary damage, gastrointestinal dysregulation, myocardial infarction, neuromuscular syndromes, neuropsychiatric disorders, pulmonary damage, renal failure, stroke, and vascular endothelial dysregulation [19]. The most commonly observed symptoms include fatigue, shortness of breath, joint pain, brain fog, anxiety and depression that affect patients’ quality of life and return to daily activities and work, causing global harm to people’s health, wellbeing, and livelihood [1,3,16,20,21]. Still, little is known about the pathophysiology of the disease, but available data implicate the multifactorial nature of COVID-19, including viral persistence, infection-triggered autoimmunity and reactivation of latent viruses, and inflammation-triggered chronic organ damage, which might explain the different phenotypes exhibited by PASC patients [1,22].

PASC diagnosis and management remains generally challenging, despite some progress made in understanding the multifaceted nature of this condition. Recent empirical evidence supports SARS-CoV-2 vaccination’s protective role against PASC development and in breakthrough infection, similarly to what has been observed with antiviral therapy [23]. Furthermore, a significant number of individuals have shown improvement of pre-existing post-COVID symptoms after vaccination [24,25,26].

Our study aims to evaluate relevant factors to PASC development in patients hospitalized in our COVID Unit and to evaluate the characteristics of patients who experience PASC symptoms.

## 2. Materials and Methods

### 2.1. Study Design and Setting

We performed a monocentric, observational, retrospective study involving a COVID-19 Unit in Naples, Italy. All patients were adults (≥18 years) and discharged alive following hospitalization with a diagnosis of SARS-CoV-2 infection confirmed by RT-PCR on a naso-oropharyngeal swab; the retrospective enrollment period ranges from 28 February 2020 to 30 April 2022. We included all patients who performed at least one follow-up visit at our post-COVID outpatient clinic, after at least 3 months from discharge. Exclusion criteria included minority age, lack of clinical data, refused or did not attend the outpatient clinic after at least 3 months from discharge, and/or lack of informed consent. All of the patients enrolled in the present study were symptomatic for COVID-19 at hospital admission. No study protocol or guidelines regarding the criteria of hospitalization were shared among the centers involved in the study, and the patients were hospitalized following the decision of physicians of each center.

All demographic and clinical data of patients with SARS-CoV-2 infection enrolled in the cohort were collected in an electronic database.

All patients admitted to our COVID Unit and discharged alive at home were prescribed a first outpatient visit approximately one month after discharge to assess the consistency of their symptoms. During the outpatient visit, the patient underwent a medical examination; in cases of persisting symptoms, we explored various possible explanations. In the absence of a definite diagnosis, these symptoms were defined as COVID-19-related. After each medical visit, based on the severity of the symptoms, the subsequent check-up date was scheduled, never exceeding 30 days.

The study was approved by the Ethics Committee of the University of Campania L. Vanvitelli, Naples (n°10877/2020). All procedures performed in this study were in accordance with the ethics standards of the institutional and/or national research committee and with the 2024 Helsinki declaration or comparable ethics standards. Informed consent was obtained from all participants included in the study.

### 2.2. Definitions

The microbiological diagnosis of SARS-CoV-2 infection was defined as a positive RT-PCR test on a naso-oropharyngeal swab. All the samples were processed using the same RT-PCR kit, Bosphore V3 (Anatolia Genework, Istanbul, Turkey, https://covid-19-diagnostics.jrc.ec.europa.eu/devices/detail/31 (accessed 15 May 2025)).

At hospitalization, characteristics of acute infection, demographic and comorbidity data were assessed including gender, age, vaccination for SARS-CoV-2, doses of vaccine received for SARS-CoV-2, active or previous smoker, number of comorbidity, presence or absence of hypertension, cardiovascular disease, atrial fibrillation, obesity, hypercholesterolemia, type 2 diabetes mellitus, chronic obstructive pulmonary disease, bronchiectasis, asthma, obstructive sleep apnea syndrome, chronic kidney disease, malignant tumor, fever, cough, dyspnea, throat ache, conjunctivitis, asthenia, myalgia, headache, ageusia/anosmia, and gastrointestinal symptoms. During hospitalization relevant clinical data were collected including presence or absence of pneumonia, time to a negative swab, remdesivir therapy during hospitalization, monoclonal antibodies treatment during hospitalization, steroids therapy during hospitalization, oxygen therapy during hospitalization, and the method of administration of oxygen therapy. At each follow-up visit at outpatient clinics after discharge, data including the presence or the absence of fever, respiratory symptoms, headache, gastrointestinal symptoms, neuropsychiatric symptoms, fatigue, myalgia, and cardiovascular symptoms were investigated and, after exclusion of known causes, were included in the electronic database.

Patients who experienced persistent clinical manifestations or the development of new symptoms 3 months after the initial SARS-CoV-2 infection, with these symptoms lasting for at least 2 months with no other explanation, were defined as having PASC [3]. Patients who were diagnosed with PASC and performed at least two visits at our outpatient clinic were stratified accordingly their clinical status, defined as follows: symptoms worsening (clinical worsening or appearance of new symptoms likely linked to PASC), clinical stability (the numeric stability of the different clinical conditions described across follow-up visits), or symptoms improving (numeric reduction of symptoms likely linked to PASC). The first group included patients who experienced worsening or stability of symptoms, and the latter included patients who experienced improvement of symptoms. During admission to our COVID-19 ward, patients’ blood oxygenation was assessed by arterial blood-gas analysis and monitored throughout their stay by pulse oximetry, according to clinical needs. During follow-up, at the outpatient clinics, patients underwent only pulse-oximetry evaluation.

### 2.3. Statistical Analysis

For the descriptive analysis, categorical variables are presented as absolute numbers and their relative frequencies. Continuous variables are summarized as mean and standard deviation or as median and interquartile range (Q1–Q3). We performed a comparison of patients with different time to a negative swab (days) using Pearson chi-square or Fisher’s exact test for categorical variables and Student’s *t* test or Mann–Whitney test for continuous variables. Odds ratios were calculated using binomial logistic regression; these analyses were performed only for parameters that were clinically relevant or that appeared statistically significant at univariate analysis. Analyses were performed by STATA (version 16) [27].

## 3. Results

During the study period, 429 patients with a diagnosis of SARS-CoV-2 infection were discharged alive from our Unit after recovery from COVID-19. Of this population, 224 patients performed at least one follow-up admission to our outpatient clinic after 12 weeks from discharge. Of these, 110 patients were defined as PASC, while PASC was ruled out for the remaining 134 (Figure 1).

Table 1 includes demographic and clinical characteristics at admission and in-hospital clinical evolution of the 224 patients grouped by the presence or absence of PASC. The 110 patients with PASC were more frequently female (43.6% vs. 31.3%, *p* = 0.048), but there were no substantial differences when considering age, vaccination status (including number of doses received), and a positive or previous smoking status (Table 1). Considering the presence of co-pathologies, only type 2 mellitus diabetes was more frequently observed in PASC patients (25.5% vs. 13%, *p* = 0.013); for the remaining variables, there were no statistical differences regarding comorbidities, either in the number or the type (Table 1). Considering the clinical characteristics of the acute phase of COVID-19, no difference between groups was observed except for sore throat and headache, which were more prevalent in PASC population (8.9% vs. 2.5%, *p* = 0.041 and 15.8% vs. 5%, *p* = 0.007, respectively) (Table 1). Interestingly, the patients treated with antiviral therapies during COVID-19 hospitalization were less likely to develop long-COVID (18.7% vs. 9.4%, *p* = 0.047) (Table 1).

At the multivariable analysis, female gender (OR 1.967, 95%CI: 1.086–3.561), the presence of type 2 diabetes (OR 2.382, 95%CI: 1.158–4.899) and headache at admission (OR 5.562, 95%CI: 1.755–17.626) were factors independently associated with PASC (Table 2).

Of the 110 patients with long-COVID, 46 (41.9%) patients performed at least two sequential admissions in our outpatient clinic; they were divided into two groups, the first including the 16 patients who experienced a reduction or a resolution of symptoms related to COVID-19 and the second including all 30 patients who experienced a worsening or a stability of symptoms (Table 3 and Table 4, Figure 2). Considering demographic and clinical variables during acute phase of COVID-19, only a smoking history or status (42.3% vs. 7.7%, *p* = 0.042) and the presence of myalgia (27.6% vs. 0%, *p*= 0.039) were more frequently observed in the group of patients with stable or worsening symptoms (Table 3).

Specifically, of the 16 patients who reported a clinical improvement, 12 (75%) reported no symptoms at the last observation, while 4 (25%) reported a reduction of symptoms; in the group comprising the 30 patients who reported stability or worsening of symptoms, 14 (46.7%) reported a worsening of symptoms and 16 (53.3%) declared a stability of symptoms during the follow-up (Table 4).

As expected, when considering the symptoms reported during follow-up, patients with worsening or stable disease had a higher cumulative number of symptoms than patients with clinical improvement (2 vs. 1, *p* = 0.0001) and a higher percentage of neuropsychiatric symptoms (53.3% vs. 6.3%, *p* = 0.0001) (Table 4). No statistical differences were found when analyzing symptoms at first observation, but at last observation, patients with worsening or stable symptoms more frequently reported neuropsychiatric symptoms (53.3% vs. 6.3%, *p* = 0.0001) and respiratory symptoms (46.7% vs. 6.3%, *p* = 0.007) (Table 3 and Table 4).

## 4. Discussion

This study aimed to identify relevant factors for PASC development in a cohort of patients hospitalized with SARS-CoV-2 infection in our COVID Unit. Our results suggest that some demographic, clinical, and in-hospital factors may influence the likelihood of developing PASC, although many of these mechanisms remain incompletely understood.

Notably, we observed a higher prevalence of PASC in female patients (43.6% vs. 31.3%, *p* = 0.048), which aligns with prior studies identifying female sex as a significant risk factor for PASC development [8,16,28]. Differently, the role of age is debated in the literature, in some cases emerging as a risk factor for PASC [29], while in others, such as in our study, it does not impact the appearance of PASC [8,28]. Among the comorbidities, Type 2 diabetes mellitus was more frequent in PASC patients (25.5% vs. 13%, *p* = 0.013), indicating a potential role of metabolic conditions in prolonged recovery [30]. In contrast, cardiovascular disease, hypertension, and chronic pulmonary conditions did not show a significant association with PASC, in contrast with previous studies [28,31,32]. Vaccination status and smoking history/status did not significantly impact the likelihood of developing PASC, although this could be attributable to the relatively small sample size and warrants further exploration, especially considering that previous studies showed a protective effect of vaccination against PASC [24,25,26]. As for acute-phase symptoms, sore throat and headache were more common in subjects who developed PASC (8.9% vs. 2.5%, *p* = 0.041, and 15.8% vs. 5%, *p* = 0.007, respectively). This supports the emerging evidence that specific acute symptoms may predict a protracted recovery [31]. However, other early indicators, relative to the acute COVID-19 phase, including the development of pneumonia and the need for oxygen therapy, were not significantly associated with PASC in our study, diverging from prior reports linking the severity of the initial infection to the development of post-acute sequelae [8,28,31,32]. The multivariable analysis, performed to analyse the predictors of PASC, confirms that headache at admission, type 2 diabetes, and female sex were independent predictors of PASC. A controversial factor is the role of antiviral treatment: in the present study, previous treatment with remdesivir during hospitalization did not result as an independent protective factor for PASC. This is not in line with previous studies, where antiviral therapy seems to serve a protective role [23]; however, it is still necessary to consider that some patients may not have received remdesivir either because they were hospitalized before its market approval or because they did not meet the prescribing criteria.

During follow-up, considering patients who performed at least two admissions to our outpatients clinic (46 patients), 16 subjects showed improvement in PASC symptoms (later reporting, during last visit, the disappearance of at least one symptom reported at the first visit), while 30 showed worsening or stable symptoms linked to PASC. Patients who showed stable or worsening symptoms linked to PASC during the follow-up more frequently reported neuropsychiatric (53.3% vs. 6.3%, *p* = 0.0001) and respiratory symptoms (46.7% vs. 6.3%, *p* = 0.007) during the last observation. These findings are consistent with existing literature indicating that fatigue, brain fog, anxiety, and dyspnea are among the most persistent and debilitating symptoms of PASC sometimes associated with a neurocognitive impairment [33,34]. The cumulative symptom burden was also significantly higher in patients whose conditions either worsened or remained stable, underscoring the complex nature of PASC and its profound impact on quality of life [35].

Although this study did not directly investigate the mechanisms underlying PASC, our results support the hypothesis that it is a multifactorial condition, potentially involving immune dysregulation/autoimmunity, viral persistence, and autoimmunity inflammation [36]. It is, however, important to point out that the study does not include in its analysis the variants that emerged during the different waves, which could still influence PASC development and exhibit distinct symptomatic characteristics, nor the presence or the absence of previous SARS-CoV-2 infection. The absence of standardized diagnostic criteria and treatment strategies remains a critical challenge, highlighting the need for further research into the pathophysiology of PASC and the development of standardized management protocols [37]. One of the strengths of this study is the inclusion of a well-defined cohort of patients with confirmed SARS-CoV-2 infection, all of whom received standardized follow-up care. However, the retrospective design, the sole enrolment of hospitalized patients, the exclusion of outpatients with asymptomatic or mild disease, and the relatively small sample size limit the generalizability of our findings. Additionally, the lack of objective biomarkers for PASC complicates the identification of reliable defining and prognostic factors; future research should aim to include larger cohorts and investigate emerging biomarkers to predict longer-term outcomes [38].

In conclusion, this study underscores the complexity of PASC, identifying female sex, Type 2 diabetes, and certain acute COVID-19 symptoms as potential risk factors. From our study, although it was conducted on a small patient population, patients who experienced persistent or worsening symptoms of PASC over time were more likely to present with neuropsychiatric and respiratory symptoms compared to those reporting clinical improvement. Furthermore, these patients also reported myalgias during the initial stages of the infection. PASC continues to pose a substantial public health challenge, and ongoing efforts are essential to better understand its underlying mechanisms and improve patient outcomes.

## Figures and Tables

**Figure 1 biomedicines-13-01414-f001:**
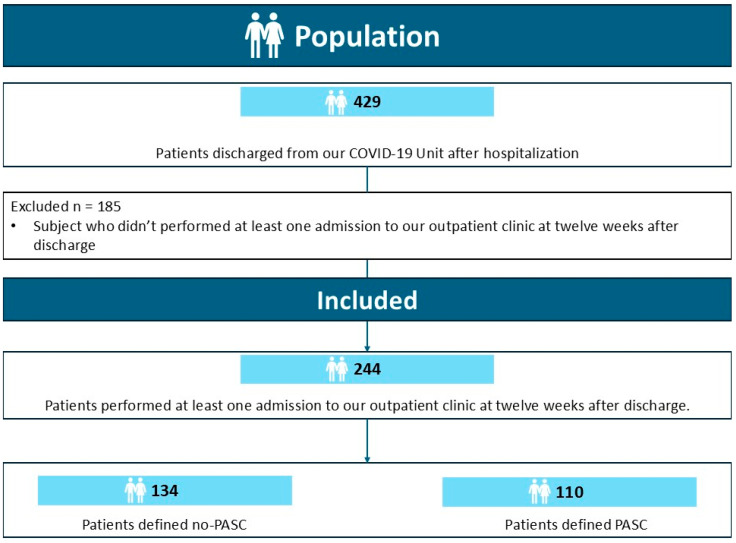
Flow chart representing the enrollment of patients in the study.

**Figure 2 biomedicines-13-01414-f002:**
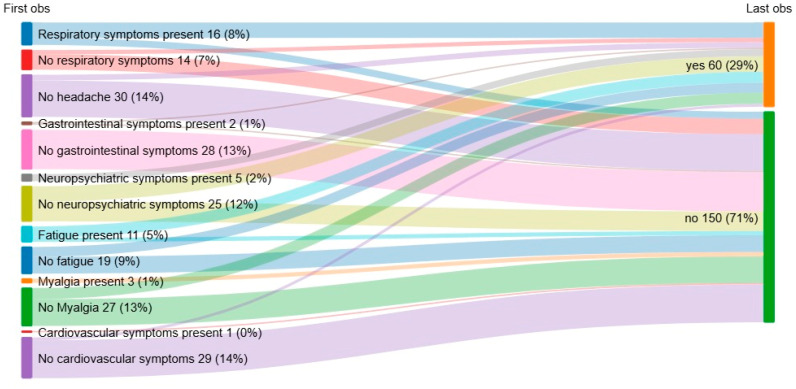
Sankey diagram showing changes in patients with stable or worsening of symptoms during follow-up considering first and last observation (obs).

**Table 1 biomedicines-13-01414-t001:** Demographic, clinical, and in-hospital admission of patients grouped by the presence or absence of PASC.

	No-PASC 134 (55%)	PASC110 (45%)	*p* Value
Sex, female N (%)	42 (31.3)	48 (43.6)	**0.048**
Age, median (IQR)	61 (52–72)	61 (52–69)	0.594
Over 65 years old, N (%)	53 (39.6%)	36 (32.7)	0.288
Vaccination, N (%)	65 (84.4%)	59 (90.8%)	0.257
Doses of vaccine received			
Partial, N (%)	8 (12.3%)	6 (10%)	0.799
Complete primary series, N (%)	32 (49.2%)	33 (55%)	
Booster, N (%)	25 (38.5%)	21 (35%)	
Co-pathologies			
Active or previous smoker, N (%); N (%)	9 (9.2%);29 (29.6%)	2 (2.4%)21 (25.6%)	0.113
At least one comorbidity, N (%)	113 (84.3%)	96 (87.3%)	0.514
Hypertension, N (%)	75 (57.3%)	63 (57.3%)	0.997
Cardiovascular disease, N (%)	23 (17.6%)	21 (19.1%)	0.759
Atrial fibrillation, N (%)	11 (8.4%)	7 (6.4%)	0.550
Obesity, N (%)	8 (6.1%)	8 (7.3%)	0.717
Hypercholesterolemia, N (%)	18 (13.7%)	25 (22.7%)	0.070
Type 2 diabetes mellitus, N (%)	17 (13%)	28 (25.5%)	**0.013**
COPD, bronchiectasis, asthma, OSAS, N (%)	13 (9.9%)	12 (10.9%)	0.803
Pulmonary fibrosis, N (%)	2 (1.5%)	2 (1.8%)	1.000
CKD, N (%)	3 (2.3%)	3 (2.7%)	1.000
Malignant tumor, N (%)	2 (1.5%)	3 (2.7%)	0.664
Autoimmune disease, N (%)	15 (11.5%)	13 (11.8%)	0.929
Cirrhosis, N (%)	2 (1.5%)	1 (0.9%)	1.000
Dementia, N (%)	1 (0.8%)	2 (1.8%)	0.593
Characteristics of acute infection			
Pneumonia, N (%)	126 (95.5%)	106 (96.4%)	0.723
Fever, N (%)	103 (85.1%)	82 (81.2%)	0.433
Cough, N (%)	56 (46.7%)	44 (43.6%)	0.644
Dyspnea, N (%)	53 (43.8%)	43 (42.6%)	0.854
Throat ache, N (%)	3 (2.5%)	9 (8.9%)	**0.041**
Conjunctivitis, N (%)	2 (1.7%)	2 (2%)	1.000
Asthenia, N (%)	23 (19%)	21 (20.8%)	0.740
Myalgia, N (%)	16 (13.2%)	19 (18.8%)	0.255
Headache, N (%)	6 (5%)	16 (15.8%)	**0.007**
Ageusia/anosmia, N (%)	8 (6.6%)	7 (6.9%)	0.925
GI symptoms, N (%)	10 (8.3%)	16 (15.8%)	0.080
Time to a negative swab, median [IQR]	17 (12–23)	16 (12–21)	0.113
Previous Remdesivir therapy, N (%)	23 (18.7%)	10 (9.4%)	**0.047**
Previous mAbs, N (%)	6 (4.5%)	4 (3.6%)	0.742
At least one antiviral treatment for SARS-CoV-2	29 (22.30%)	14 (13.21%)	0.071
Previous steroids, N (%)	108 (85.7%)	91 (85.8%)	0.977
Previous O_2_ therapy, N (%)	108 (85.7%)	89 (83.2%)	0.593
Mode of O_2_ therapy			0.897
VM or NC, N (%)	33 (33%)	24 (28.2%)	
HFNC, N (%)	30 (30%)	29 (34.1%)	
CPAP or NIV, N (%)	36 (36%)	31 (36.5%)	
ETI, N (%)	1 (1%)	1 (1.2%)	

Footnotes: COPD, chronic obstructive pulmonary disease; O_2_: oxygen; OSAS, obstructive sleep apnea syndrome; CKD, chronic kidney disease; GI, gastro-intestinal; mAbs, monoclonal antibodies; VM, Venturi Mask; NC, nasal cannulae; HFNC, high-flow nasal cannulae; CPAP, continuous positive airway pressure; NIV, non-invasive ventilation; ETI, endotracheal intubation. The bold highlight statistical significance in univariate or multivariable analysis

**Table 2 biomedicines-13-01414-t002:** Multivariable logistic regression analysis assessing the factor associated with PASC.

	OR (Confidence Interval 95%)	*p*
Female gender	2.010 (1.096–3.687)	**0.024**
Age, median	0.996 (0.972–1.020)	0.719
Type 2 diabetes	2.388 (1.161–4.912)	**0.018**
Headache at admission	5.555 (1.754–17.597)	**0.004**
Previous Remdesivir therapy	0.503 (0.217–1.166)	0.109

The bold highlight statistical significance in univariate or multivariable analysis.

**Table 3 biomedicines-13-01414-t003:** Demographic, clinical, and in-hospital data of PASC patients grouped by clinical evolution during hospitalization and at the first visit to outpatient clinic after three months from discharge.

	Patients Improving Symptoms 16 (34.8%)	Stable or Worsening Patients 30 (65.2%)	*p* Value
Gender, male N (%)	13 (81.3%)	19 (63.3)	0.316
Age, median (IQR)	58 (52–69)	58 (54–71)	0.644
Over 65 years old, N (%)	5 (31.3%)	9 (30.0%)	0.930
Vaccination, N (%)	12 (92.3%)	22 (95.7%)	1.000
Doses of vaccine received			
Partial, N(%)	2 (16.7%)	0 (0%)	0.102
Complete primary series, N(%)	7 (58.3%)	13 (56.5%)	
Booster, N(%)	3 (25%%)	10 (43.5%)	
Active or previous smoker, N (%); N (%)	2 (15.4%)	11 (42.3%)	0.152
At least one comorbidity, N (%)	15 (93.8%)	26 (86.7%)	0.645
Hypertension, N (%)	9 (56.3%)	20 (66.7%)	0.534
Cardiovascular disease, N (%)	2 (12.5%)	6 (20%)	0.694
Atrial fibrillation, N (%)	1 (6.3%)	1 (3.3%)	1.000
Obesity, N (%)	1 (6.3%)	3 (10%)	1.000
Hypercholesterolemia, N (%)	5 (31.3%)	3 (10%)	0.105
Type 2 diabetes mellitus, N (%)	4 (25%)	0 (0%)	0.720
COPD, bronchiectasis, asthma, OSAS, N (%)	1 (6.3%)	4 (13.3%)	0.645
CKD, N (%)	0 (0%)	1 (3.3%)	1.000
Malignant tumor, N (%)	1 (6.3%)	2 (6.6%)	1.000
Autoimmune disease, N (%)	1 (6.3%)	1 (3.3%)	1.000
Cirrhosis, N (%)	0 (0%)	0 (0%)	-
Dementia, N (%)	0 (0%)	0 (0%)	-
Characteristics of acute infection			
Fever, N (%)	10 (71.4%)	30 (89.7%)	0.190
Cough, N (%)	8 (57.1%)	10 (34.5%)	0.158
Dyspnea, N (%)	6 (42.9%)	10 (34.5%)	0.594
Throat ache, N (%)	1 (7.1%)	0 (0%)	0.326
Conjunctivitis, N (%)	0 (0%)	0 (0%)	-
Asthenia, N (%)	3 (21.4%)	6 (20.7%)	1.000
Myalgia, N (%)	0 (0%)	8 (27.6%)	**0.039**
Headache, N (%)	2 (14.3%)	6 (20.7%)	1.000
Ageusia/anosmia, N (%)	1 (7.1%)	2 (6.9%)	1.000
GI symptoms, N (%)	2 (14.3%)	2 (6.9%)	0.585
Time to negative swab, median [IQR]	16 (13–19)	17 (13–22)	0.945
Previous Remdesivir therapy, N (%)	2 (12.5%)	2 (7.1%)	0.614
Previous mAbs, N (%)	1 (6.3%)	0 (0%)	0.348
Previous steroids, N (%)	13 (81.3%)	28 (96.6%)	0.137
Previous O2 therapy, N (%)	11 (78.6%)	28 (96.3%)	0.094
Mode of O_2_ therapy			0.809
VM or NC, N (%)	3 (27.3%)	6 (23.1%)	
HFNC, N (%)	5 (45.5%)	10 (38.5%)	
CPAP or NIV, N (%)	3 (27.3%)	10 (38.5%)	
ETI, N (%)	0 (0%)	0 (0%)	
Symptoms at first observation in outpatient clinic			
Cumulative symptoms at first observation, median [IQR]	1 (1–2)	1 (1–2)	0.879
Fever at first observation, N (%)	0 (0%)	1 (3.3%)	1.000
Respiratory symptoms at first observation, N (%)	7 (43.8%)	16 (53.3%)	0.536
Headache at first observation, N (%)	0 (0%)	0 (0%)	-
GI symptoms at first observation, N (%)	1 (6.3%)	2 (6.7%)	1.000
Neuropsychiatric symptoms at first observation, N (%)	1 (6.3%)	5 (16.7%)	0.649
Fatigue at first observation, N (%)	9 (56.3%)	11 (36.7%)	0.229
Myalgia at first observation, N (%)	3 (18.8%)	3 (10%)	0.405
Cardiovascular symptoms at first observation, N (%)	1 (6.3%)	1 (3.3%)	1.000
Fever at last observation, N (%)	0 (0%)	2 (6.7%)	0.536

Footnotes: COPD, chronic obstructive pulmonary disease; OSAS, obstructive sleep apnea disease; CKD, chronic kidney disease; GI, gastro-intestinal; mAbs, monoclonal antibodies; VM, Venturi Mask; NC, nasal cannulae; HFNC, high-flow nasal cannulae; CPAP, continuous positive airway pressure; NIV, non-invasive ventilation; ETI, endotracheal tube intubation. The bold highlight statistical significance in univariate or multivariable analysis.

**Table 4 biomedicines-13-01414-t004:** Evolution of reported PASC symptoms of patients during follow-up grouped considering improvement or worsening of symptoms during follow-up.

	Patients Improving Symptoms 16 (34.8%)	Stable or Worsening Patients 30 (65.2%)	*p* Value
Symptoms described in outpatient clinic during FU			
Cumulative symptoms during FU, median [IQR]	1 (1–1)	2 (2–3)	**0.0001**
Fever during FU, N (%)	0 (0%)	2 (6.7%)	0.536
Respiratory symptoms during FU, N (%)	6 (37.5%)	19 (63.3%)	0.094
Headache during FU, N (%)	0 (0%)	4 (13.3%)	0.282
GI symptoms during FU, N (%)	1 (6.3%)	2 (6.7%)	1.000
Neuropsychiatric symptoms during FU, N (%)	1 (6.3%)	16 (53.3%)	**0.001**
Fatigue during FU, N (%)	7 (43.8%)	17 (56.7%)	0.404
Myalgia during FU, N (%)	4 (25%)	8 (26.7%)	1.000
Cardiovascular symptoms during FU, N (%)	1 (6.3%)	3 (10%)	1.000
Symptoms at last observation in outpatient clinic			
Respiratory symptoms at last observation, N (%)	1 (6.3%)	14 (46.7%)	**0.007**
Headache at last observation, N (%)	0 (0%)	4 (13.3%)	0.282
GI symptoms at last observation, N (%)	1 (6.3%)	1 (3.3%)	1.000
Neuropsychiatric symptoms at last observation, N (%)	0 (0%)	16 (53.3%)	**0.001**
Fatigue at last observation, N (%)	2 (12.5%)	15 (50%)	0.023
Myalgia at last observation, N (%)	1 (6.3%)	8 (26.7%)	0.132
Cardiovascular symptoms at last observation, N (%)	0 (0%)	2 (6.7%)	0.536

Footnotes: FU, follow-up; GI, gastro-intestinal. The bold highlight statistical significance in univariate or multivariable analysis.

## Data Availability

The original contributions presented in this study are included in the article. Further inquiries can be directed to the corresponding author.

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
