# Peer review of "Prevalence, Evolution and Prognostic Factors of PASC in a Cohort of Patients Discharged from a COVID Unit"

_biomedicines, 2025, doi:10.3390/biomedicines13061414_

Round 1

Reviewer 1 Report

Comments and Suggestions for Authors

This manuscript analyses the prevalence and prognostic factors associated with post-acute sequelae of SARS-CoV-2 infection (PASC) or long COVID using retrospective data from a single-centre cohort in Naples, Italy. The study is of high quality and is generally well written. However, I have several comments that may help improve the scientific clarity and overall value of the manuscript.

Comments.
1. Age group and additional variables in the multivariable regression model:
Did you consider including age groups (e.g., <60 vs. ≥60 or <65 vs. ≥65) in the multivariable logistic regression analysis?

Several studies have reported a higher likelihood of PASC among older individuals. This may be due to immunosenescence, age-related vulnerability to severe acute COVID-19, or biological susceptibility to prolonged symptoms. Your study may provide additional evidence supporting this association.
Reference: https://www.ncbi.nlm.nih.gov/pmc/articles/PMC10922032/

In Table 1, the “Age” variable does not appear to differ significantly between groups based on the rank test (presumably a Mann–Whitney U test). However, it is important to note that univariate or bivariate statistical tests (e.g., Mann–Whitney U, t-test, Chi-square, Fisher's exact test) only assess differences for a single variable between groups.
In contrast, multivariable regression evaluates the contribution of multiple variables simultaneously.

Therefore, a non-significant or significant result in univariate/bivariate testing or even in univariable regression does not rule out the variable as a potentially significant predictor in a multivariable model.

I suggest trying extended analysis, including age groupings or other relevant variables in the multivariable model, based on clinical or epidemiological assumptions. 
If it works, this may enhance the strength and impact of your findings.

2. Ageusia and anosmia:
Were ageusia and anosmia recorded separately during data collection? 
If so, I recommend reporting them as distinct symptoms in the results table, rather than combining them.
In previous studies, anosmia has typically occurred more frequently than ageusia or concurrent ageusia and anosmia.

3. Prior SARS-CoV-2 infection:
Did any participants experience previous COVID-19 infections prior to the one evaluated in this study? 
Reinfection or multiple-time infections could influence both the acute clinical presentation, disease severity and PASC.
Consider clarifying whether prior infection history was assessed (by interview, questionnaire or other methods) and whether this was accounted for in your analysis.

4. Pulse oximeter data during acute infection:
Did the study collect oxygen saturation data via pulse oximetry during the acute phase?
If so, suggest clarifying the outcomes that may reveal more interesting results, and clarify how this data was collected (e.g., outpatient visits, COVID-19 ward admission).

5. SARS-CoV-2 prominent variant wave considerations:
The enrollment period spans a broad timeframe. It is covering the ancestral-related strain, Alpha, Delta, to multiple Omicron subvariants (e.g., BA.1, BA.2, BA.5, BA.2.75).
Each prominent variant wave is associated with differing clinical manifestations, transmissibility, severity, and potentially different risks for PASC. Moreover, changes in cumulative infection rates and vaccine coverage over time may also introduce confounding.

Suggest clarifying whether the timing of infection (by variant wave) was considered or whether outcomes could be influenced by variant-specific effects. 
If such data are unavailable, consider briefly discussing this as a potential limitation.

6. COVID-19 vaccination details:
Vaccination plays an important role in reducing both COVID-19 severity and PASC risk. However, its protective effect depends on the number of doses and the time interval between the most recent dose and infection.

Consider clarify:
- The number of doses received (e.g., partial, complete primary series, booster or 1, 2 ≥3), not presented as "median with IQR".
- The time between the last vaccine dose and infection (e.g., 14 days–1 month, >1–3 months, >3 months), if available.
This could support an interpretation of the vaccination’s role in reducing PASC as seen in the literature.

7. Remdesivir use and timeframe:
Remdesivir received full FDA approval on 22 October 2020 (with EMA and AIFA approvals likely following).
Participants infected prior to the availability of remdesivir may not have received this treatment, which may have influenced the severity of their acute illness and subsequent PASC risk.

Please consider carefully discussing this point, especially if your data show differences in outcomes by infection date or treatment status.

8. Statistical software:
In the Methods section, please specify a certain version of STATA used in the analysis to enhance reproducibility (e.g. version xx.y.z).

Author Response

Dear Editor,

We re-submit our paper “PREVALENCE, EVOLUTION AND PROGNOSTIC FACTORS OF LONG COVID IN A COHORT OF PATIENTS DISCHARGED FROM A COVID UNIT.” (manuscript n°: biomedicines- 3635521), modified according to the suggestions of the Reviewers.

ANSWERS TO THE COMMENTS OF THE REVIEWER 1

Point 1.  Age group and additional variables in the multivariable regression model: Did you consider including age groups (e.g., <60 vs. ≥60 or <65 vs. ≥65) in the multivariable logistic regression analysis? Several studies have reported a higher likelihood of PASC among older individuals. This may be due to immunosenescence, age-related vulnerability to severe acute COVID-19, or biological susceptibility to prolonged symptoms. Your study may provide additional evidence supporting this association.
Reference: https://www.ncbi.nlm.nih.gov/pmc/articles/PMC10922032/ In Table 1, the “Age” variable does not appear to differ significantly between groups based on the rank test (presumably a Mann–Whitney U test). However, it is important to note that univariate or bivariate statistical tests (e.g., Mann–Whitney U, t-test, Chi-square, Fisher's exact test) only assess differences for a single variable between groups. In contrast, multivariable regression evaluates the contribution of multiple variables simultaneously. Therefore, a non-significant or significant result in univariate/bivariate testing or even in univariable regression does not rule out the variable as a potentially significant predictor in a multivariable model. I suggest trying extended analysis, including age groupings or other relevant variables in the multivariable model, based on clinical or epidemiological assumptions. 
If it works, this may enhance the strength and impact of your findings.

Answer: We thank you for the suggestions. We improve the manuscript accordingly.

Point 2. Ageusia and anosmia: Were ageusia and anosmia recorded separately during data collection?  If so, I recommend reporting them as distinct symptoms in the results table, rather than combining them. In previous studies, anosmia has typically occurred more frequently than ageusia or concurrent ageusia and anosmia.

Answer: Unfortunately, in our data collection we don’t record the symptoms separately.

Point 3. Prior SARS-CoV-2 infection: Did any participants experience previous COVID-19 infections prior to the one evaluated in this study?  Reinfection or multiple-time infections could influence both the acute clinical presentation, disease severity and PASC. Consider clarifying whether prior infection history was assessed (by interview, questionnaire or other methods) and whether this was accounted for in your analysis.

Answer: we modified the manuscript according to the suggestion of the reviewer.

Point 4. Pulse oximeter data during acute infection: Did the study collect oxygen saturation data via pulse oximetry during the acute phase? If so, suggest clarifying the outcomes that may reveal more interesting results, and clarify how this data was collected (e.g., outpatient visits, COVID-19 ward admission).

Answer: Following the suggestion of the reviewer, we include it in the method section.

Point 5. SARS-CoV-2 prominent variant wave considerations: The enrolment period spans a broad timeframe. It is covering the ancestral-related strain, Alpha, Delta, to multiple Omicron subvariants (e.g., BA.1, BA.2, BA.5, BA.2.75).
Each prominent variant wave is associated with differing clinical manifestations, transmissibility, severity, and potentially different risks for PASC. Moreover, changes in cumulative infection rates and vaccine coverage over time may also introduce confounding. Suggest clarifying whether the timing of infection (by variant wave) was considered or whether outcomes could be influenced by variant-specific effects. If such data are unavailable, consider briefly discussing this as a potential limitation.

Answer: We thanks you for the suggestion, but we didn’t have these data. We underlined this limit in the new manuscript.

Point 6. COVID-19 vaccination details: Vaccination plays an important role in reducing both COVID-19 severity and PASC risk. However, its protective effect depends on the number of doses and the time interval between the most recent dose and infection. Consider clarify:
- The number of doses received (e.g., partial, complete primary series, booster or 1, 2 ≥3), not presented as "median with IQR".
- The time between the last vaccine dose and infection (e.g., 14 days–1 month, >1–3 months, >3 months), if available.
This could support an interpretation of the vaccination’s role in reducing PASC as seen in the literature.

Answer: We modified the manuscript as suggested by the reviewer.

Point 7. Remdesivir use and timeframe: Remdesivir received full FDA approval on 22 October 2020 (with EMA and AIFA approvals likely following).
Participants infected prior to the availability of remdesivir may not have received this treatment, which may have influenced the severity of their acute illness and subsequent PASC risk. Please consider carefully discussing this point, especially if your data show differences in outcomes by infection date or treatment status.

Answer: We modified the discussions according to the suggestion of the reviewer.

Point 8. Statistical software: In the Methods section, please specify a certain version of STATA used in the analysis to enhance reproducibility (e.g. version xx.y.z).

Answer: As suggested by the reviewer, we included it in the method section of the new manuscript.

ANSWERS TO THE COMMENTS OF THE REVIEWER 2

Point 1. Terminology – "Long COVID"

The term “Long COVID” is being used with increasing caution in academic circles. As suggested in recent work by Professor Ziyad Al-Aly, a leading expert in the field, adopting more contemporary and precise terminology may enhance the manuscript's clarity and alignment with current research. His publications may serve as valuable references in this regard.

Answer: We thank you for the suggestion, we modified the text accordingly.

Point 2. Clarification of Data Source

Although I am actively engaged in Long COVID research, the description of the dataset used in this study is somewhat unclear. To improve transparency and reproducibility, please consider providing more detailed information regarding patient inclusion criteria, data collection methods, and linkage processes.

Answer: Following the suggestion of the reviewer, we improved the methods.

Point 3. Ethical Considerations – Declaration of Helsinki

While the Declaration of Helsinki was initially adopted in 1964, it has undergone multiple revisions, the most recent being in October 2024. If applicable, I recommend that the ethics statement be updated to reflect the latest version of the Declaration.

Answer: As required by the reviewer, we included this point.

Point 4. Methods Section

The methods section currently lacks sufficient detail. I encourage the authors to strengthen this section by referencing recent high-quality studies that have examined the risks and sequelae of Long COVID. Relevant literature (e.g., PMID: 40275542, 40251935) may help substantiate the study's rationale and methodology.

Answer: We modified the text according to the suggestion of the reviewer.

Point 5. Figure 1 – Flow Diagram

The flowchart presented in Figure 1 appears overly simplified. A more comprehensive and professionally formatted diagram could improve both the clarity of the study design and the visual presentation.

Answer: Following the suggestion of the reviewer, we modified the figure 1 

Point 6. Terminology – “Non-Long COVID” vs. “Long COVID”

The dichotomous classification of patients as “non-Long COVID” and “Long COVID” may be overly reductive or inaccurate. It may be beneficial to use terms that reflect clinical consensus or current literature, such as “Post-Acute Sequelae of SARS-CoV-2 infection (PASC).”

Answer: We thank you for the suggestion. We modified accordingly.

Point 7. Reporting Odds Ratios and Confidence Intervals

In the tables, it may be more effective to present odds ratios alongside their 95% confidence intervals in a single column rather than in separate cells. This format can improve readability and streamline interpretation.

Answer: As suggested, we modified the tables.

Point 8. Reporting of p-values

Please ensure consistent reporting of p-values with appropriate precision throughout the manuscript. For instance, values less than 0.001 should be reported as p < 0.001, and exact values such as 1 should be reported as p = 1.000, maintaining uniformity to three decimal places.

Answer: We modified the text according to the suggestion.

We thank the Reviewers for helping us to improve our paper.

We hope that the paper is now worthy of publication in “Biomedicines”

Best regards,

Prof Nicola Coppola

Reviewer 2 Report

Comments and Suggestions for Authors

Thank you for the opportunity to review the manuscript. Given the clinical and public health significance of this topic, the manuscript demonstrates considerable potential and may be suitable for publication pending several key revisions.

  1. Terminology – "Long COVID"
    The term “Long COVID” is being used with increasing caution in academic circles. As suggested in recent work by Professor Ziyad Al-Aly, a leading expert in the field, adopting more contemporary and precise terminology may enhance the manuscript's clarity and alignment with current research. His publications may serve as valuable references in this regard.
  2. Clarification of Data Source
    Although I am actively engaged in Long COVID research, the description of the dataset used in this study is somewhat unclear. To improve transparency and reproducibility, please consider providing more detailed information regarding patient inclusion criteria, data collection methods, and linkage processes.
  3. Ethical Considerations – Declaration of Helsinki
    While the Declaration of Helsinki was initially adopted in 1964, it has undergone multiple revisions, the most recent being in October 2024. If applicable, I recommend that the ethics statement be updated to reflect the latest version of the Declaration.
  4. Methods Section
    The methods section currently lacks sufficient detail. I encourage the authors to strengthen this section by referencing recent high-quality studies that have examined the risks and sequelae of Long COVID. Relevant literature (e.g., PMID: 40275542, 40251935) may help substantiate the study's rationale and methodology.
  5. Figure 1 – Flow Diagram
    The flowchart presented in Figure 1 appears overly simplified. A more comprehensive and professionally formatted diagram could improve both the clarity of the study design and the visual presentation.
  6. Terminology – “Non-Long COVID” vs. “Long COVID”
    The dichotomous classification of patients as “non-Long COVID” and “Long COVID” may be overly reductive or inaccurate. It may be beneficial to use terms that reflect clinical consensus or current literature, such as “Post-Acute Sequelae of SARS-CoV-2 infection (PASC).”
  7. Reporting Odds Ratios and Confidence Intervals
    In the tables, it may be more effective to present odds ratios alongside their 95% confidence intervals in a single column rather than in separate cells. This format can improve readability and streamline interpretation.
  8. Reporting of p-values
    Please ensure consistent reporting of p-values with appropriate precision throughout the manuscript. For instance, values less than 0.001 should be reported as p < 0.001, and exact values such as 1 should be reported as p = 1.000, maintaining uniformity to three decimal places.

Author Response

Dear Editor,

We re-submit our paper “PREVALENCE, EVOLUTION AND PROGNOSTIC FACTORS OF LONG COVID IN A COHORT OF PATIENTS DISCHARGED FROM A COVID UNIT.(manuscript n°: biomedicines- 3635521), modified according to the suggestions of the Reviewers.

ANSWERS TO THE COMMENTS OF THE REVIEWER 1

Point 1.  Age group and additional variables in the multivariable regression model: Did you consider including age groups (e.g., <60 vs. ≥60 or <65 vs. ≥65) in the multivariable logistic regression analysis? Several studies have reported a higher likelihood of PASC among older individuals. This may be due to immunosenescence, age-related vulnerability to severe acute COVID-19, or biological susceptibility to prolonged symptoms. Your study may provide additional evidence supporting this association.
Reference: https://www.ncbi.nlm.nih.gov/pmc/articles/PMC10922032/ In Table 1, the “Age” variable does not appear to differ significantly between groups based on the rank test (presumably a Mann–Whitney U test). However, it is important to note that univariate or bivariate statistical tests (e.g., Mann–Whitney U, t-test, Chi-square, Fisher's exact test) only assess differences for a single variable between groups. In contrast, multivariable regression evaluates the contribution of multiple variables simultaneously. Therefore, a non-significant or significant result in univariate/bivariate testing or even in univariable regression does not rule out the variable as a potentially significant predictor in a multivariable model. I suggest trying extended analysis, including age groupings or other relevant variables in the multivariable model, based on clinical or epidemiological assumptions. 
If it works, this may enhance the strength and impact of your findings.

Answer: We thank you for the suggestions. We improve the manuscript accordingly.

Point 2. Ageusia and anosmia: Were ageusia and anosmia recorded separately during data collection?  If so, I recommend reporting them as distinct symptoms in the results table, rather than combining them. In previous studies, anosmia has typically occurred more frequently than ageusia or concurrent ageusia and anosmia.

Answer: Unfortunately, in our data collection we don’t record the symptoms separately.

Point 3. Prior SARS-CoV-2 infection: Did any participants experience previous COVID-19 infections prior to the one evaluated in this study?  Reinfection or multiple-time infections could influence both the acute clinical presentation, disease severity and PASC. Consider clarifying whether prior infection history was assessed (by interview, questionnaire or other methods) and whether this was accounted for in your analysis.

Answer: we modified the manuscript according to the suggestion of the reviewer.

Point 4. Pulse oximeter data during acute infection: Did the study collect oxygen saturation data via pulse oximetry during the acute phase? If so, suggest clarifying the outcomes that may reveal more interesting results, and clarify how this data was collected (e.g., outpatient visits, COVID-19 ward admission).

Answer: Following the suggestion of the reviewer, we include it in the method section.

Point 5. SARS-CoV-2 prominent variant wave considerations: The enrolment period spans a broad timeframe. It is covering the ancestral-related strain, Alpha, Delta, to multiple Omicron subvariants (e.g., BA.1, BA.2, BA.5, BA.2.75).
Each prominent variant wave is associated with differing clinical manifestations, transmissibility, severity, and potentially different risks for PASC. Moreover, changes in cumulative infection rates and vaccine coverage over time may also introduce confounding. Suggest clarifying whether the timing of infection (by variant wave) was considered or whether outcomes could be influenced by variant-specific effects. If such data are unavailable, consider briefly discussing this as a potential limitation.

Answer: We thanks you for the suggestion, but we didn’t have these data. We underlined this limit in the new manuscript.

Point 6. COVID-19 vaccination details: Vaccination plays an important role in reducing both COVID-19 severity and PASC risk. However, its protective effect depends on the number of doses and the time interval between the most recent dose and infection. Consider clarify:
- The number of doses received (e.g., partial, complete primary series, booster or 1, 2 ≥3), not presented as "median with IQR".
- The time between the last vaccine dose and infection (e.g., 14 days–1 month, >1–3 months, >3 months), if available.
This could support an interpretation of the vaccination’s role in reducing PASC as seen in the literature.

Answer: We modified the manuscript as suggested by the reviewer.

Point 7. Remdesivir use and timeframe: Remdesivir received full FDA approval on 22 October 2020 (with EMA and AIFA approvals likely following).
Participants infected prior to the availability of remdesivir may not have received this treatment, which may have influenced the severity of their acute illness and subsequent PASC risk. Please consider carefully discussing this point, especially if your data show differences in outcomes by infection date or treatment status.

Answer: We modified the discussions according to the suggestion of the reviewer.

Point 8. Statistical software: In the Methods section, please specify a certain version of STATA used in the analysis to enhance reproducibility (e.g. version xx.y.z).

Answer: As suggested by the reviewer, we included it in the method section of the new manuscript.

ANSWERS TO THE COMMENTS OF THE REVIEWER 2

Point 1. Terminology – "Long COVID"

The term “Long COVID” is being used with increasing caution in academic circles. As suggested in recent work by Professor Ziyad Al-Aly, a leading expert in the field, adopting more contemporary and precise terminology may enhance the manuscript's clarity and alignment with current research. His publications may serve as valuable references in this regard.

Answer: We thank you for the suggestion, we modified the text accordingly.

Point 2. Clarification of Data Source

Although I am actively engaged in Long COVID research, the description of the dataset used in this study is somewhat unclear. To improve transparency and reproducibility, please consider providing more detailed information regarding patient inclusion criteria, data collection methods, and linkage processes.

Answer: Following the suggestion of the reviewer, we improved the methods.

Point 3. Ethical Considerations – Declaration of Helsinki

While the Declaration of Helsinki was initially adopted in 1964, it has undergone multiple revisions, the most recent being in October 2024. If applicable, I recommend that the ethics statement be updated to reflect the latest version of the Declaration.

Answer: As required by the reviewer, we included this point.

Point 4. Methods Section

The methods section currently lacks sufficient detail. I encourage the authors to strengthen this section by referencing recent high-quality studies that have examined the risks and sequelae of Long COVID. Relevant literature (e.g., PMID: 40275542, 40251935) may help substantiate the study's rationale and methodology.

Answer: We modified the text according to the suggestion of the reviewer.

Point 5. Figure 1 – Flow Diagram

The flowchart presented in Figure 1 appears overly simplified. A more comprehensive and professionally formatted diagram could improve both the clarity of the study design and the visual presentation.

Answer: Following the suggestion of the reviewer, we modified the figure 1

Point 6. Terminology – “Non-Long COVID” vs. “Long COVID”

The dichotomous classification of patients as “non-Long COVID” and “Long COVID” may be overly reductive or inaccurate. It may be beneficial to use terms that reflect clinical consensus or current literature, such as “Post-Acute Sequelae of SARS-CoV-2 infection (PASC).”

Answer: We thank you for the suggestion. We modified accordingly.

Point 7. Reporting Odds Ratios and Confidence Intervals

In the tables, it may be more effective to present odds ratios alongside their 95% confidence intervals in a single column rather than in separate cells. This format can improve readability and streamline interpretation.

Answer: As suggested, we modified the tables.

Point 8. Reporting of p-values

Please ensure consistent reporting of p-values with appropriate precision throughout the manuscript. For instance, values less than 0.001 should be reported as p < 0.001, and exact values such as 1 should be reported as p = 1.000, maintaining uniformity to three decimal places.

Answer: We modified the text according to the suggestion.

We thank the Reviewers for helping us to improve our paper.

We hope that the paper is now worthy of publication in “Biomedicines”

Best regards,

Prof Nicola Coppola

Round 2

Reviewer 1 Report

Comments and Suggestions for Authors

Thank you for thoroughly addressing the concerns I raised in your previous submission.